# Identification of common non-coding variants at 1p22 that are functional for non-syndromic orofacial clefting

Huan Liu[1,2], Elizabeth J. Leslie[3], Jenna C. Carlson[4], Terri H. Beaty[5], Mary L. Marazita[3,6], Andrew C. Lidral[7] & Robert A. Cornell[1]

Genome-wide association studies (GWAS) do not distinguish between single nucleotide polymorphisms (SNPs) that are causal and those that are merely in linkage-disequilibrium with causal mutations. Here we describe a versatile, functional pipeline and apply it to SNPs at 1p22, a locus identified in several GWAS for non-syndromic cleft lip with or without cleft palate (NS CL/P). First we amplified DNA elements containing the ten most-highly risk-associated SNPs and tested their enhancer activity *in vitro*, identifying three SNPs with allele-dependent effects on such activity. We then used *in vivo* reporter assays to test the tissue-specificity of these enhancers, chromatin configuration capture to test enhancer–promoter interactions, and genome editing *in vitro* to show allele-specific effects on ARHGAP29 expression and cell migration. Our results further indicate that two SNPs affect binding of CL/P-associated transcription factors, and one affects chromatin configuration. These results translate risk into potential mechanisms of pathogenesis.

[1] Department of Anatomy and Cell Biology, College of Medicine, University of Iowa, Iowa City, Iowa 52242, USA. [2] State Key Laboratory Breeding Base of Basic Science of Stomatology (Hubei-MOST) and Key Laboratory for Oral Biomedicine of Ministry of Education, School and Hospital of Stomatology, Wuhan University, Wuhan, Hubei 430079, China. [3] Center for Craniofacial and Dental Genetics, Department of Oral Biology, School of Dental Medicine, University of Pittsburgh, Pittsburgh, Pennsylvania 15219, USA. [4] Department of Biostatistics, Graduate School of Public Health, University of Pittsburgh, Pittsburgh, Pennsylvania 15261, USA. [5] Department of Epidemiology, Bloomberg School of Public Health, Johns Hopkins University, Baltimore, Maryland 21205, USA. [6] Department of Human Genetics, Graduate School of Public Health and Clinical and Translational Science Institute, School of Medicine, University of Pittsburgh, Pittsburgh, Pennsylvania 15219, USA. [7] Department of Orthodontics, College of Dentistry, University of Iowa, Iowa City, Iowa 52246, USA. Correspondence and requests for materials should be addressed to R.A.C. (email: robert-cornell@uiowa.edu).

C left lip with or without cleft palate (CL/P) affects about 1 in 700 live births in the US, and has both genetic and environmental underpinnings[1]. About 30% of cases of CL/P are syndromic and may be inherited in Mendelian fashion, whereas the remaining cases are non-syndromic (NS) and appear to be controlled by multiple genes[2] and environmental factors[2]. Collectively, eight independent genome-wide association studies (GWAS)[3–10], a linkage study[11], meta-analyses of GWAS[12–14] and several replication studies (for example, ref. 13) provide statistical support achieving genome-wide significance for associations between single nucleotide polymorphism (SNP) markers and risk for NS CL/P. The GWAS approach has been unusually successful in identifying loci in which variation contributes significantly to risk for NS CL/P (ref. 8), in comparison to its degree of success for other complex diseases[15].

There are several challenges to translating statistical associations revealed by GWAS into an understanding of the biological causes of NS CL/P. First, only a subset of SNPs in linkage disequilibrium (LD) with the lead GWAS SNP are likely directly pathogenic (that is, functional), but it is nearly impossible to distinguish these with statistical approaches[16,17]. Second, virtually all SNPs associated with NS CL/P lie within non-coding DNA segments, as is the case for most other complex diseases. SNPs within non-coding DNA are presumed, in most cases, to disrupt cis-regulatory modules. Because the connection between the sequence and function of enhancers is currently poorly understood[18], it is not immediately clear how a given change of a single base pair could alter the function of any given enhancer. Third, the identity of the 'risk gene' (that is, the one whose expression level influences risk for NS CL/P) is not necessarily obvious, because enhancers can lie at an arbitrary distance from the genes they regulate, skipping over intervening genes or lying within the introns of other genes[19,20]. Finally, once a functional SNP is identified, the mechanism by which a single-base-pair difference influences enhancer activity must be examined experimentally.

To address these challenges, we applied an experimental pipeline to the 1p22 region associated with NS CL/P (refs 3,7,8,21,22). This locus was one of 13 regions selected for resequencing in NS CL/P case-parent trios. This effort discovered multiple SNPs strongly associated with NS CL/P that were in strong LD with the most significant SNP found in a GWAS, all within the introns of ABCA4 (ref. 22). Craniofacial enhancers have been identified within the introns of ABCA4 (ref. 23), but ABCA4 gene is not a good candidate for the risk gene because, first, in mice, expression of Abca4 is not detectable in the developing lip or palate[3]; second, in mice homozygous for targeted loss-of-function mutations in Abca4, no defects in craniofacial structure were reported[24]; and, finally, in humans, coding mutations in ABCA4 that cause profound defects in retinal function (that is, Stargardt's disease 1 (refs 25–27)) do not appear to cause CL/P. By contrast, a neighbouring gene, ARHGAP29 is expressed in mouse palate and lip, its expression depends on IRF6, and coding variants in ARHGAP29 are associated with CL/P (ref. 28).

We hypothesized that functional SNPs in 1p22 lie within enhancers that drive ARHGAP29 expression in one or more oral tissue, and the risk alleles at these SNPs decrease ARHGAP29 expression by altering the binding affinities of specific transcription factors. In vitro enhancer studies, chromosome-conformation capture, zebrafish-based in vivo enhancer assays, plus genome editing all provide evidence that three SNPs along with two haplotypes are likely to be functional. Finally, chromatin immunoprecipitation analyses indicate that the risk-associated alleles of these SNPs affect the activity of ARHGAP29 enhancers by altering binding of specific transcription factors, and in one

case disrupts the interaction of the enhancer with the ARHGAP29 promoter.

## Results

**Reporter assays reveal SNPs that affect enhancer activity.** We prioritized ten SNPs from the 1p22 locus for our experimental pipeline. All were strongly associated with risk for NS CL/P in Asian case-parent trios used in our resequencing project after correcting for multiple testing and all passed quality control filters[22]. The prioritized SNPs are listed in Supplementary Table 1 and depicted in Fig. 1a and Supplementary Fig. 1. We reasoned that functional SNPs will reside in enhancers, while SNPs with no functional effect but in LD with a causal SNP (sometimes called rider or hitch-hiking SNPs) may not. We first tested whether DNA elements containing these SNPs have enhancer activity in oral epithelium or palate mesenchyme cells, the two major cell types contributing to palate tissue. Eight of the ten SNPs lie within chromatin regions expressing marks indicative of enhancer activity in one or more of the 127 cell lines evaluated in the Roadmap Epigenomics project[29] (Fig. 1a, middle track). For these SNPs, we amplified DNA elements that approximately matched the boundaries of such marks and contained one or more risk-associated SNP. For the two SNPs that did not fall into such chromatin regions, we amplified approximately 1 kb of DNA centred on each SNP (Fig. 1a). In total we cloned six enhancer candidates (E1–E6). The resequencing project revealed just two alleles each at all ten SNPs; the risk allele was the major allele at two SNPs and the minor allele at eight SNPs (Supplementary Table 1). As necessary, in each enhancer candidate we converted the risk-allele to the non-risk allele by site-directed mutagenesis[22]. We also amplified four negative-control segments (C1–C4) from nearby chromatin that lacked marks indicative of enhancer or promoter function in the 127 cell lines (positions shown in Fig. 1a).

We engineered E1–E6 and C1–C4 into a standard firefly luciferase vector, electroporated each reporter construct, together with a plasmid that drives constitutive expression of renilla luciferase (transfection control), into a human fetal oral epithelial cell line (GMSM-K)[30] and primary human embryonic palate mesenchymal (HEPM)[31] cells, and monitored luciferase activity 72 h later. C1–C4 had similar low-level enhancer activity, whose average we defined as the baseline (separately for the two cell types). The activity of five of the six enhancer candidates in oral-epithelium cells was at least five-fold higher than baseline (Fig. 1c); the exception was E4, one of the two elements for which evidence of chromatin marks was lacking. Of the five with strong enhancer in oral epithelium, four also had enhancer activity in palate mesenchyme cells at least two-fold above baseline, although in each case much lower than in oral-epithelium cells. Based on these results, nine of ten SNPs passed the first criterion of residing within an active enhancer in an oral-epithelium cell line and or palate mesenchyme cells.

We next hypothesized that functional SNPs would have allele-dependent effects on the activity level of the enhancers containing them. To this end, we engineered the SNPs in each element from the non-risk allele to the risk-associated allele, and then repeated the in vitro luciferase reporter assay in both cell types (Fig. 2). For elements that contain more than one SNP (that is, E3 and E6), we altered one SNP at a time. We then prioritized SNPs based on whether they had allele-dependent effects on enhancer activity. For instance, the level of enhancer activity of E1 in oral epithelium cell lines and oral mesenchyme cells was independent of the allele of its resident SNP-of-interest, rs581244 (Fig. 2a, Supplementary Table 2), indicating this SNP is unlikely to be directly functional. By contrast, the enhancer activity of E2 in oral epithelium cells was

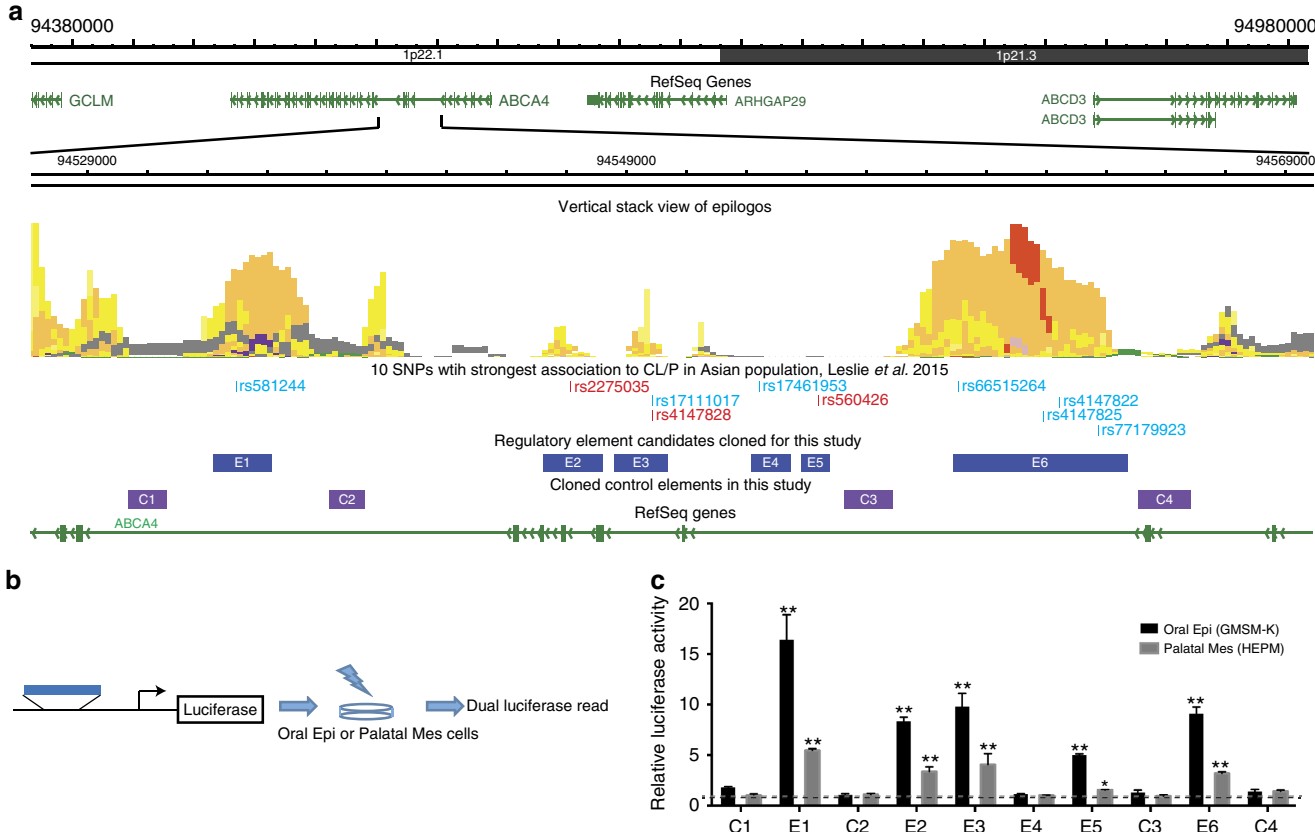

**Figure 1 | *In vitro* enhancer screening for *cis*-regulatory elements harbouring Top 10 NS CL/P-associated SNPs.** (**a**) Summary of all the elements cloned and tested in 1p22 NS CL/P-associated locus in this study. (Vertical stack view of epilogos was generated from a chromatin state model based on imputed data provided by NIH Roadmap Epigenomics Consortium; http://egg2.wustl.edu/roadmap/web_portal). Shown are the locations of ten SNPs with strongest association to CL/P in the Asian population (data from ref. 22). *Red font*, functional SNPs validated in this study, *Blue font*, non-functional SNPs, based on the results of this study. (**b**) Schematic illustration of the *in vitro* enhancer screening strategy using the cloned elements. (**c**) Relative luciferase activities of all the cloned candidate regulatory elements and control elements in GMSM-K (human oral keratinocyte cell line) and HEPM (human embryonic palatal mesenchymal cells). Black dash line indicates the average relative luciferase activity of control elements in GMSM-K, and grey dash line indicates the average relative luciferase activity of control elements in HEPM. $n = 3$ for each group, data represent means ± s.d., t-test, *$P < 0.05$, **$P < 0.01$ significant difference compared with average relative luciferase activity of control elements in respective cell type.

30% lower when its resident SNP (rs2275035) harboured the risk versus the non-risk allele (Fig. 2a) ($P = 0.0001$ by Student's t-test), supporting the candidacy of this SNP for being functional. Of note, E2 enhancer activity in palate mesenchyme cells was independent of the allele at rs2275035, favouring oral epithelium as the risk-relevant tissue of expression (Fig. 2b) ($P = 0.2049$ by Student's t-test). Based on allele-dependent effects in *in vitro* enhancer assays, rs2275035 (in E2), rs4147828 (in E3), rs560426 (in E5) and rs66515264 (in E6) emerged as good candidates for functional SNPs (Fig. 2, data summarized in Supplementary Table 2). Among these, we chose to pursue the three that decrease enhancer activity, that is, SNPs rs2275035 (in E2), rs4147828 (in E3) and rs560426 (in E5).

***In vivo* enhancer tests reveal tissue specificity.** The tissue specificity of enhancers containing these three putatively functional SNPs can reveal the basis of disease mechanisms, so we employed *in vivo* enhancer assays in zebrafish. To this end, we engineered the five elements passing the *in vitro* assay (that is, all except E4) into GFP reporter vectors, using the minimal promoter of either *cfos* or *gata2*. Each construct was injected into more than 200 embryos, and GFP expression was monitored during the second and third days of development of the F0 generation. In a subset of embryos injected with the E2-based constructs, we detected GFP expression in cranial epidermis, notochord, blood vessels, brain and trabecula (dorsal neurocranium cartilage), regardless of which minimal promoter was used. Similarly, in embryos injected with the E3-based constructs, GFP expression was detected in the same tissues; both reporter constructs yielded grossly similar patterns (Supplementary Table 3). In contrast, embryos injected with reporter constructs containing E1 or E5 did not demonstrate consistent patterns of expression, and in embryos injected with E6-based constructs, GFP was routinely detected only in neurons (Supplementary Table 3).

We raised E2 and E3-injected founders to adulthood and identified at least three that transmitted the transgene to the F1 generation. Stable transgenic F1 embryos (that is, *Tg(E2:GFP)* and *Tg(E3:GFP)* embryos) exhibited a non-mosaic pattern of GFP expression matching that of their respective founders. In *Tg(E2:GFP)* embryos at 5 h post-fertilization (h.p.f.) GFP fluorescence was not detectable, but at 30 h.p.f. it was visible in the brain and head epidermis, and by 4 days post-fertilization (d.p.f.) it was clearly present in retina, oral epithelium, and in palate mesenchyme (trabeculae and ethmoid plate) (shown in transverse sections, see Fig. 3c,e). In *Tg(E3:GFP)* embryos, similarly, at 5 h.p.f. GFP expression was not detectable, but at 30 h.p.f. it was evident in brain (Supplementary Fig. 2), and by 4 d.p.f. it was similar to that seen in *Tg(E2:GFP)* larvae although

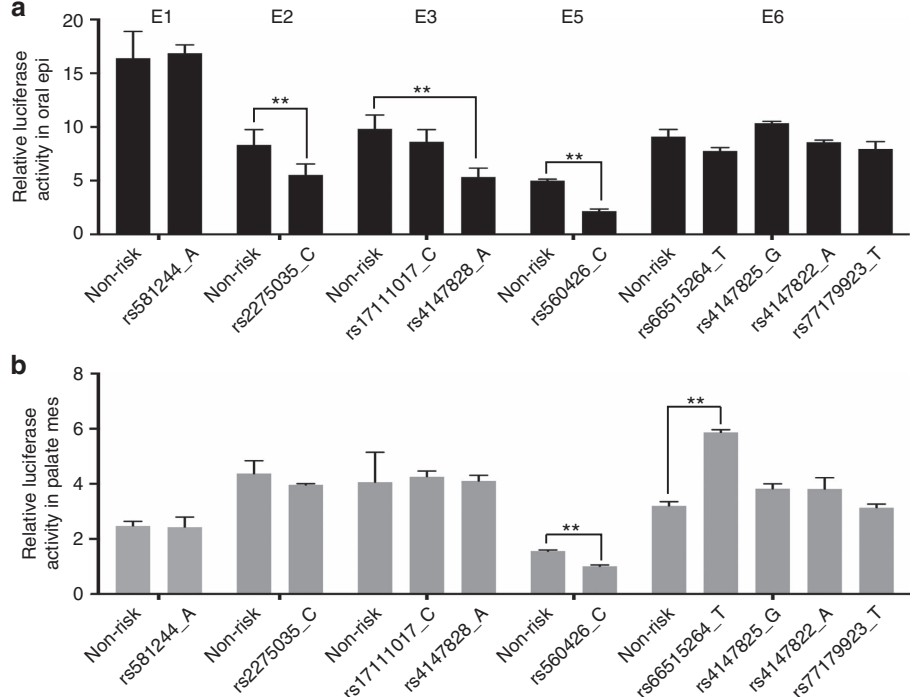

**Figure 2 | Allele-specific effects of NS CL/P-associated SNPs on the *in vitro* activity of regulatory elements in GMSM-K cells and HEPM cells. (a)** Relative luciferase activity of different elements with different alleles in GMSM-K cells. (**b**) Relative luciferase activity of different elements with different alleles in HEPM cells. $n = 3$ for each group, data represent means ± s.d., *t*-test, **$P < 0.01$ significant difference compared with enhancer activities of each element with non-risk allele in respective cell type.

brain expression was lower (Fig. 3b,d,f). We also compared the F0 GFP patterns produced by E2 and E3 harbouring risk versus non-risk alleles as indicated in the *in vitro* screen. However, we did not detect a difference in the level or spatial distribution of GFP (Supplementary Tables 4 and 5). Based on the enhancer activity of these elements *in vitro* data (Figs 1 and 2) and *in vivo* (Fig. 3), we conclude that epithelial tissues, that is, oral epithelium or potentially cranial epidermis, are the most likely to be relevant to the pathogenesis conferred by variation at these SNPs.

**3C-qPCR identifies enhancer–promoter interactions.** We next sought to identify the gene or genes whose expression is regulated by enhancers harbouring the functional SNP candidates. Enhancers rarely interact with promoters outside of the topologically associated domain (TAD) in which they reside[32,33]. TADs appear to be delimited by loci bound by CCCTC-binding factor, and, conveniently, TAD boundaries are largely consistent among cell types[32–34]. Referring to HiC results from normal human embryonic keratinocyte (NHEK) and IMR90 myofibroblast cell lines[35], we observed that the ten NS CL/P-associated SNPs of interest are located within a single TAD containing three genes: *ABCA4*, *ARHGAP29* and *ABCD3* (Fig. 4a and Supplementary Fig. 3).

To examine chromatin interactions between the NS CL/P-associated region and the promoters of these three candidate genes, we performed chromatin configuration capture (3C) in the human oral epithelium cell line and human palate mesenchyme cells. Briefly, we isolated nuclei, digested chromatin with the restriction enzyme EcoR1, and re-ligated the chromatin under dilute conditions. Next we carried out quantitative PCR (qPCR), using anchor primers at the transcription start sites (promoters) of *ABCA4*, *ARHGAP29* or *ABCD3* and bait primers in EcoR1-restriction fragments containing the various NS CL/P-associated SNPs.

The quantities of ligation product between the EcoR1 restriction fragment containing the *ARHGAP29* promoter and (a) the one containing E2, (b) the one containing E3, (c) the one containing an element homologous to a craniofacial enhancer identified by p300 ChIP-SEQ in mouse facial tissue (mouse element 435) (ref. 23) were higher than those of it and other nearby fragments (Fig. 4b).

E4 and E5 exhibited relatively lower interaction frequency with the *ARHGAP29* promoter. This was a surprising finding for E5, given the *in vitro* results presented above; it is noteworthy that the nearest EcoR1 site is over 4 kb from E5, perhaps limiting the efficiency of 3C in detecting interactions between the *ARHGAP29* promoter and E5. By contrast, ligation products for the EcoR1 restriction fragment containing the *ABCA4* and *ABCD3* promoter and all fragments in those regions were similarly low (Fig. 4c). Also, as expected, four EcoRI-digested fragments in the adjacent TAD exhibited very low interaction frequency with the promoters of *ARHGAP29* (Supplementary Fig. 3b), *ABCA4* (Supplementary Fig. 3c) and *ABCD3* (Supplementary Fig. 3d). Similar outcomes were predicted using PreSTIGE, another method for evaluating intrachromsomal interactions, in NHEKs (ref. 36). These experiments strongly support *ARHGAP29* as the NS CL/P risk gene in the 1p22 region.

The functional allele of an SNP could potentially affect the strength of an interaction between an enhancer and its corresponding promoter. We tested this possibility for rs4147828 (in E3). PCR and sequencing revealed that oral epithelium cells are heterozygous for the risk and non-risk alleles (Fig. 4c, INPUT). By contrast, in the amplicon of the 3C ligation products between the EcoRI-restriction fragments containing E3 and the *ARHGAP29* promoter, only the non-risk allele at rs4147828 was detected (Fig. 4c). This result suggests the risk allele at rs4147828 disrupts the interactions between enhancer E3 and the *ARHGAP29* promoter. It remains to be seen whether other functional SNPs have a similar effect or alter enhancer function by a distinct mechanism.

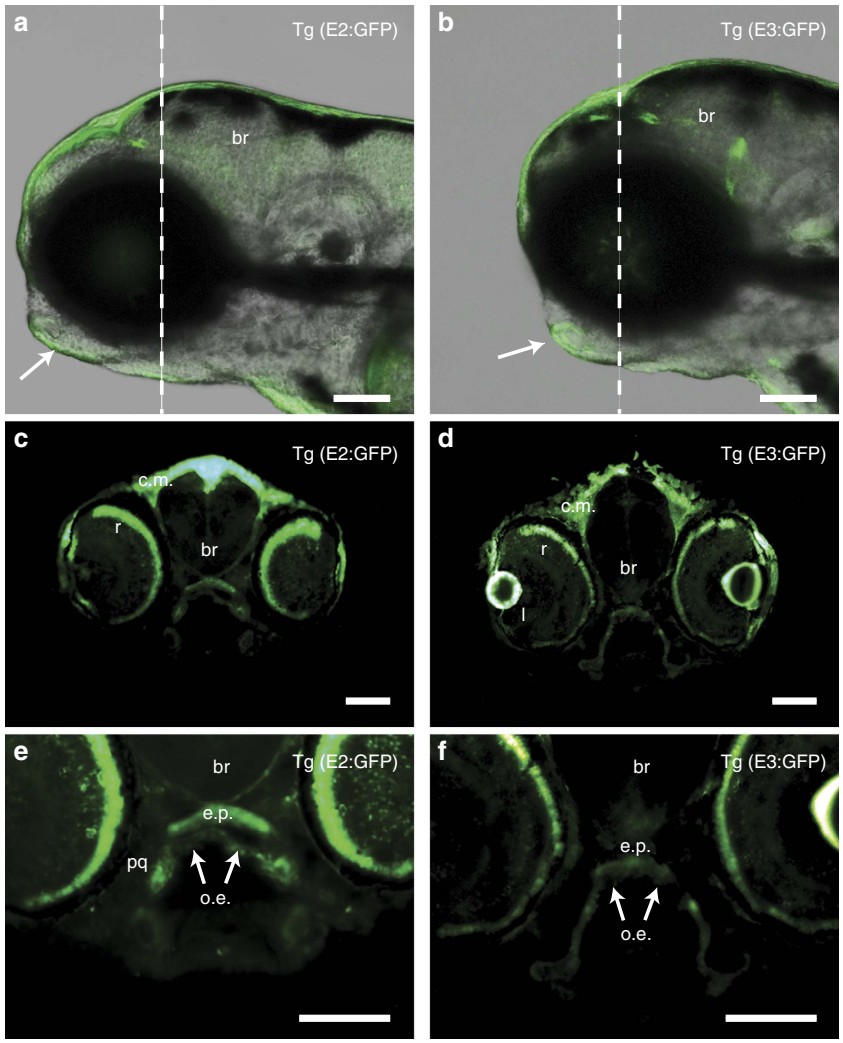

**Figure 3 | Stable transgenic zebrafish lines showing *in vivo* enhancer activities patterns of E2 and E3 in craniofacial region.** (**a**,**b**) Z-stacks from confocal imaging of live stable (F1) transgenic GFP-reporter lines for (**a**) E2 and (**b**) E3 at 4 d.p.f. (representative of F1 lines from at least three independent F0 founders). Arrows indicate GFP expression in ventral epithelium and oral epithelium. White dashed lines in **a**,**b** indicate section planes in **c**–**f**, respectively. (**c**–**f**) Transverse sections (12 μm) of stable GFP-reporter lines of E2 (**c**,**e**) and E3 (**d**,**f**) show *in vivo* activity of E2 in head epidermis, retina (r), lens (l), ethmoid plate (e.p.), oral epithelium (o.e., white arrow), brain (br) and palatoquadrate (pq). Scale bars = 100 μm.

**Genome editing shows SNPs affect expression of *ARHGAP29*.** We next tested whether deletion of E2 or E3 from the genome alters expression of *ARHGAP29* in oral epithelium cells. We transfected the oral epithelium cell line with plasmids encoding Cas9 and a guide RNA (one on either side of the enhancer), performed selection and clonal isolation, and used PCR to identify cells in which E2, E3, or both had undergone bi-allelic deletion (Fig. 5a and Supplementary Fig. 4). In cells in which either E2 (3 isolated colonies) or E3 (3 isolated colonies) was eliminated, the expression of *ARHGAP29* was significantly lower than in control cells ($P < 0.01$ by Student's *t*-test); in cells lacking both elements (three isolated colonies), *ARHGAP29* expression was less than 50% of normal levels (Fig. 5b). However, the expression of *ABCD3* remained unchanged and that of *ABCA4* was undetectable in cells of any genotype (Fig. 5b). These results confirm that both E2 and E3 are enhancers for *ARHGAP29* in this cell line.

For E5, rather than delete the entire enhancer, we used CRISPR-Cas9-mediated homology-directed repair (HDR) to delete 10 bp centred on its candidate functional SNP (rs560426), or to alter the allele of this SNP (see Methods). We identified at least five independent colonies with each engineered genotype, and used quantitative RT–PCR to monitor *ARHGAP29* expression in each. Compared to unmanipulated cells, which were heterozygous for risk and non-risk alleles, cells engineered to be homozygous for the 10-bp deletion, or to be homozygous for the risk allele, had significantly lower average *ARHGAP29* expression (10-bp deletion $P = 8.0E-06$, homozygous for risk allele, $P = 5.0E-04$, by Student's *t*-test), whereas cells engineered to be homozygous for the non-risk allele had significantly higher average *ARHGAP29* expression ($P = 2.8E-05$ by Student's *t*-test) (Fig. 5f). In contrast, expression of *ABCA4* was undetectable in all colonies tested, and *ABCD3* expression was not affected by genome editing (Supplementary Fig. 7). These results confirm that expression of *ARHGAP29* in this cell line is affected by the nucleotide at rs560426 (E5), supporting the candidacy of this SNP for being directly functional.

Next, we used HDR to engineer the alleles of rs2275035 (in E2) and rs4147828 (in E3). At both SNPs, unmanipulated GMSM-K oral epithelium cells are heterozygous for the risk and non-risk alleles (Fig. 5d). Using a two-step selection strategy, we first knocked in the desired allele in a template containing a neomycin-resistance (neo) cassette, and then removed the neo

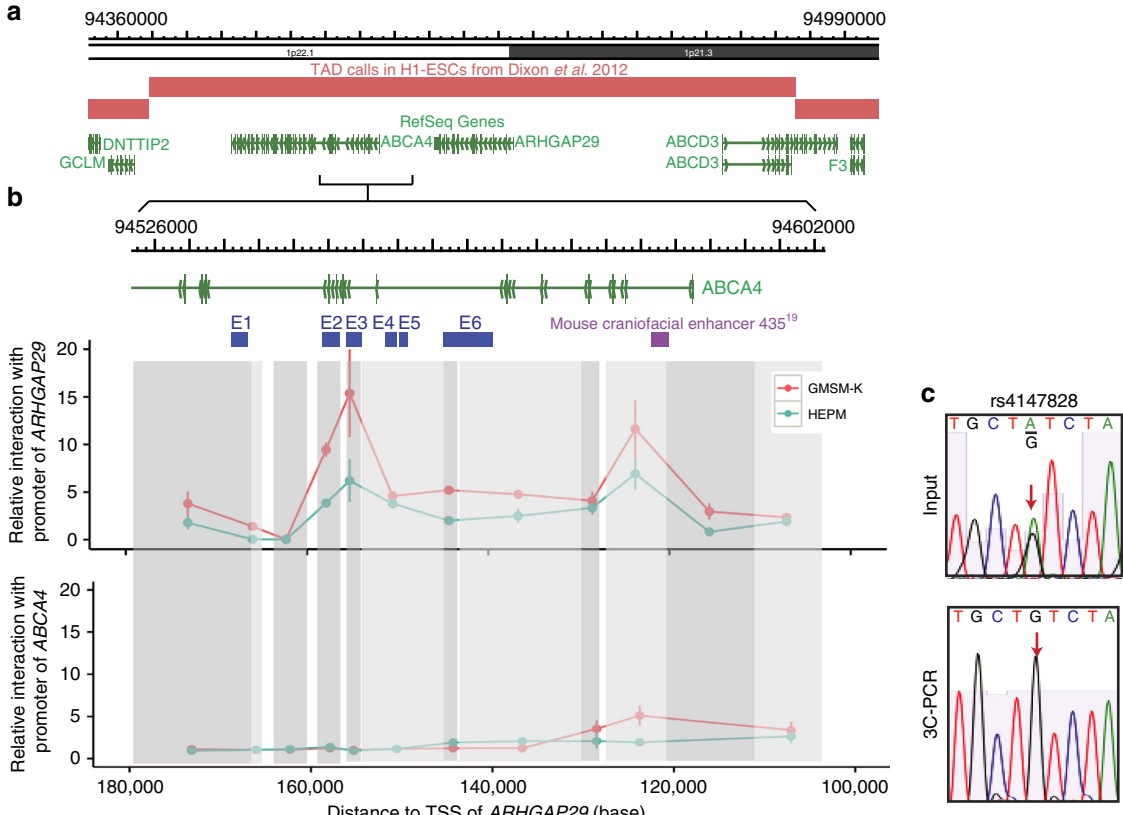

**Figure 4 | Chromatin interactions at the 1p22 NS CL/P-associated region with *ARHGAP29*.** (**a**) Topologically associated domains at the 1p22 NS CL/P-associated region showing candidate affected genes. (**b**) 3C-qPCR showing relative interactions frequency of 1p22 regions with promoter of *ARHGAP29* and *ABCA4* in GMSM-K and HEPM cells. Vertical grey boxes indicate EcoRI fragments interrogated. Purple box, craniofacial enhancer, identified in mouse genome[23] and lifted over to human genome. n = 3 for each cell type. Points represent mean relative interaction frequency, while vertical lines indicate standard deviations. (**c**) Sanger-sequencing results of 3C products showing allele-specific chromatin looping between E3 and the *ARHGAP29* promoter.

cassette, leaving behind a single loxP site of 34 base pairs (Supplementary Fig. 5a–d). As for HDR in E2, although the presence of the neo cassette modestly affected *ARHGAP29* expression, the presence of the loxP site did not (that is, *ARHGAP29* expression was not different in unedited 'T/C' cells and 'loxP-T/loxP-C' cells, $P > 0.30$ by Student's $t$-test) (Fig. 5g). In three independent isolates of colonies of each genotype, average *ARHGAP29* expression was higher in cells homozygous for the non-risk allele ($P = 0.057$ by Student's $t$-test), and significantly higher in heterozygotes ($P = 7.5E-3$ by Student's t-test), than in those homozygous for the risk allele (Fig. 5g). The results for rs4147828 were similar (Fig. 5h). Thus, the expression of *ARHGAP29* appears to be affected by the nucleotide at rs2275035 and rs4147828, supporting that they are functional SNPs.

ARHGAP29 inactivates RhoA (ref. 37) and RhoA regulates keratinocyte migration[38,39]; gain and loss-of-function experiments have shown that ARHGAP29 positively regulate oral keratinocyte migration *in vitro*, while mutant *ARHGAP29* identified in orofacial clefting patients is unstable and fails to accelerate cell migration[40]. We next asked whether the level of change of *ARHGAP29* expression mediated by the allele of a potentially functional SNP is sufficient to alter cell migratory behaviour *in vitro*. We performed a scratch assay, using three independent colonies per engineered genotype and unedited colonies. We grew each culture to confluency, made several scratches in the cell lawn and monitored cell migration for 24 h. Cells homozygous for deletion of E2, E3 or E2 + E3 closed the scratches more slowly than unedited cells (Fig. 5i,j). Moreover,

cells homozygous for risk alleles at SNPs rs2275035, rs4147828 and rs560426 migrated more slowly than those homozygous for the corresponding non-risk alleles (Fig. 5i,j and Supplementary Fig. 8). These findings indicate that variation at these SNPs is sufficient to alter the migratory behaviour of epithelial cells, presumably by altering *ARHGAP29* expression, further supporting the candidacy of these SNPs as directly functional.

**Engineered risk haplotypes affect *ARHGAP29* expression.** Having shown that the allele at single SNPs can affect enhancer activity and *ARHGAP29* expression, we next sought to evaluate the effect of specific risk haplotypes. In our published targeted resequencing results, we reported two independent signals in Asian case-parent trios at the 1p22 locus: one tagged by rs560426 (in E5) and a second haplotype tagged by rs77179923 (in E6) (ref. 22). We revisited these data to examine haplotype frequencies with SNPs within or between enhancer candidates using phased parental data. Seven SNPs were in moderate to high LD with rs560426 (peak 1) and three SNPs were in moderate to high LD with rs77179923 (peak 2) (Supplementary Fig. 9).

In peak 1, rs2275035 (E2) and rs4147828 (E3) were in strong LD ($r^2 = 0.61$, $D' = 0.98$) and, unsurprisingly, a haplotype comprised of both risk alleles (that is, the CA haplotype) was over-transmitted to NS CL/P offspring ($P = 0.0015$ by Student's $t$-test), while a haplotype comprised of the non-risk alleles of both SNPs (that is, the TG haplotype) was under-transmitted ($P = 8.00E-04$ by Student's $t$-test) (Supplementary Table 6). Phasing of the oral epithelium cells we used in this study

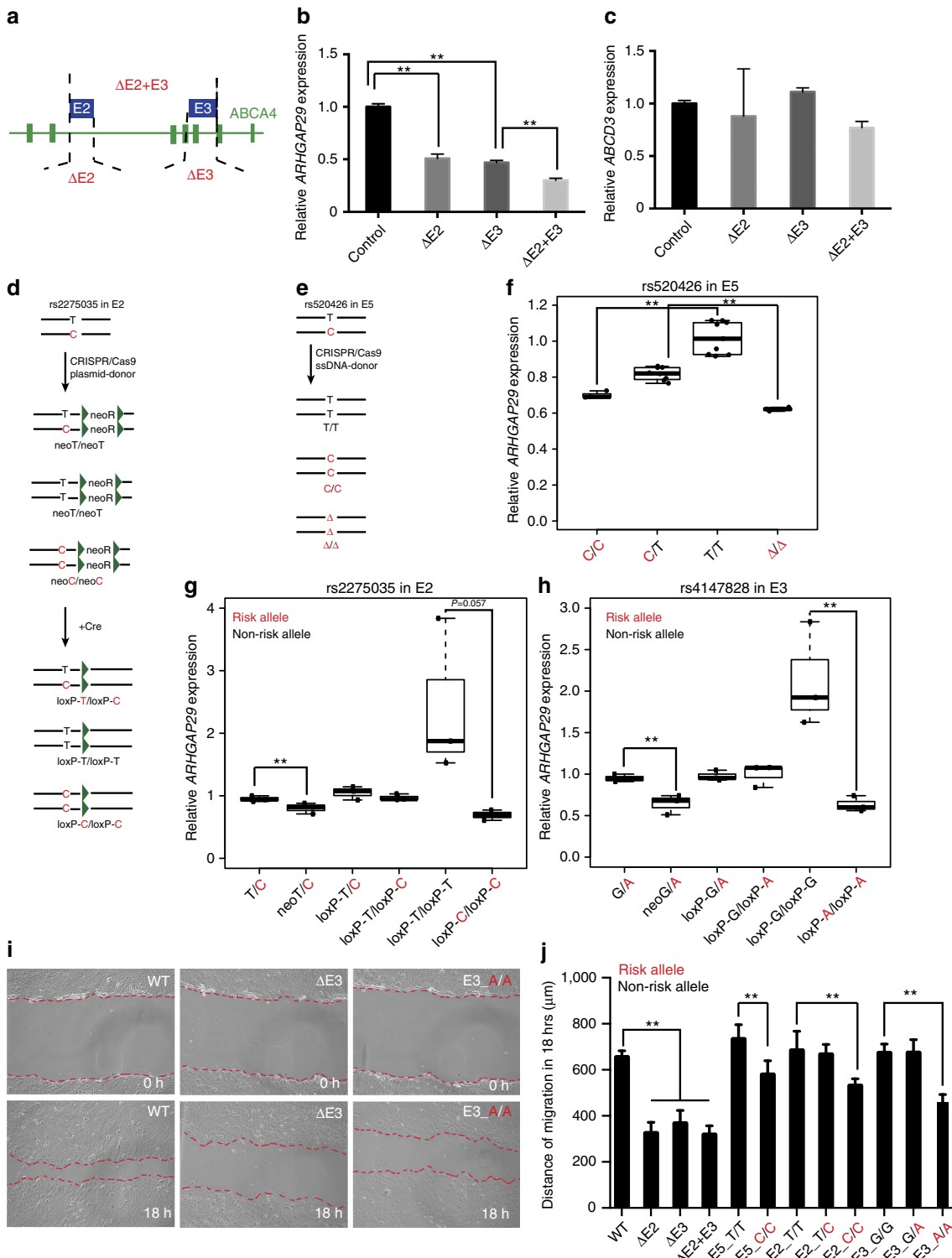

**Figure 5 | Allele-specific effects of three SNPs in 1p22 on *ARHGAP29* expression.** (**a**) Schematic illustrations of the experimental strategy to generate homozygous deletion of E2 (ΔE2) and E3 (ΔE3) respectively or both (ΔE2 + E3) in GMSM-K cells. (**b,c**) Relative expression of *ARHGAP29* and *ABCD3* in ΔE2, ΔE3 and ΔE2 + E3 cells. Each box indicates average expression of three isolated colonies (*n* = 3) with same enhancer deletion. For each colony, ARHGAP29 and ABCD3 expression were measured for three times. Data represent means ± s.d., *t*-test, **P < 0.01. (**d**) Schematic showing the HDR of rs4147828 in E3. (**e**) Schematic showing the HDR of rs560426 in E5. (**f–h**) Relative expression of *ARHGAP29* in cells with different genotypes. Each dot represents mean expression of *ARHGAP29* in one isolated colony. (**i**) Representative photos showing migration of oral epithelium cell with different genotype (unedited, rs4147828_A allele (E3_A in figure), and ΔE3) in 0 h and 18 h after scratch. (**j**) Distance of migration of oral epithelial cells with different genotypes within 18 h after scratch. (E5_C/C indicates colony with homozygous C allele of rs4147828 in E5, and so forth.) Distances between two sides of scratch of each group at each time point were measured using ImageJ. *n* = 3 for each group, data represent means ± s.d., *t*-test, *P < 0.05, **P < 0.01.

revealed two rarer haplotypes (TA/CG), meaning each chromosome carried only one risk allele. The HDR strategy we used above yielded colonies with haplotypes CA/CG (three total risk alleles) or haplotypes TG/CG, with only one risk allele (Fig. 6a). As expected, average *ARHGAP29* expression in colonies with CA/CG were significantly lower than the colonies with TG/CG and the original un-treated cells (Fig. 6b). The additive effect of combining risk alleles at both E2 and E3 was also observed in *in vitro* reporter assays where the enhancer was E2 fused to E3 (Fig. 6c).

The E3 element contained rs4147828 and rs17111017, both among the top ten SNPs, as well as three other SNPs with significant TDT *p*-values that were not among the top ten SNPs. These five SNPs appear in the following order rs75017380, rs3789419, rs17111017, rs4147828 and rs4147827 (Fig. 6d). The genotype of the unedited oral epithelium cell line is TGCGC/TGCAC (that is, heterozygous for the non-risk G allele and risk A allele at rs4147828). Analysis of phased haplotypes from the sequencing data revealed TGCGC is a non-risk haplotype, significantly under-transmitted to NS CL/P cases ($P = 7E-04$ by Student's *t*-test) (Supplementary Table 7), while TGCAC is a neutral haplotype, neither under- nor over-transmitted to cases ($P = 0.51$ by Student's *t*-test). This suggested that a particular combination of risk alleles, including the other E3 SNPs, may be required to affect *ARHGAP29* expression. To test this possibility, we used appropriate homology arms to make the cells homozygous for TGGAG, a risk haplotype, significantly under-

transmitted to NS CL/P cases ($P = 0.0004$ by Student's *t*-test) (Supplementary Table 7). In experiments discussed above targeting rs4147828 we had already generated cells homozygous for the non-risk haplotype, TGCGC/TGCGC, and for the neutral haplotype, TGCAC/TGCAC. Contrary to the suggestion mentioned above, *ARHGAP29* expression in cells homozygous for the risk-haplotype and for the neutral haplotype was not significantly different (Fig. 6e). Together these results support the idea SNP rs4147828 is functional while the other SNPs in E3 are carried along by linkage. They also imply the reason that TGGAG is a risk haplotype and TGCAC is a neutral one is merely due to information content and statistical power rather than functional effects of these other SNPs.

Finally, we tested for an association between all four SNPs with allele-specific effects: rs2275035 (in E2), rs4147828 (in E3) and rs560426 (in E5) from peak 1 and rs66515264 (in E6) from peak 2. Interestingly, the only haplotype that was significantly associated with NS CL/P was one including risk alleles at all four variants (Supplementary Table 7).

**ChIP reveals SNP allele affects transcription factor binding.** We hypothesized that the allele of functional SNP affects the binding affinity of developmental transcription factors. We used JASPAR (ref. 41) to predict binding event that would be affected by the allele of the functional SNP candidates. rs2275035 (in E2) lies within a binding site for KLF4 whose predicted quality is

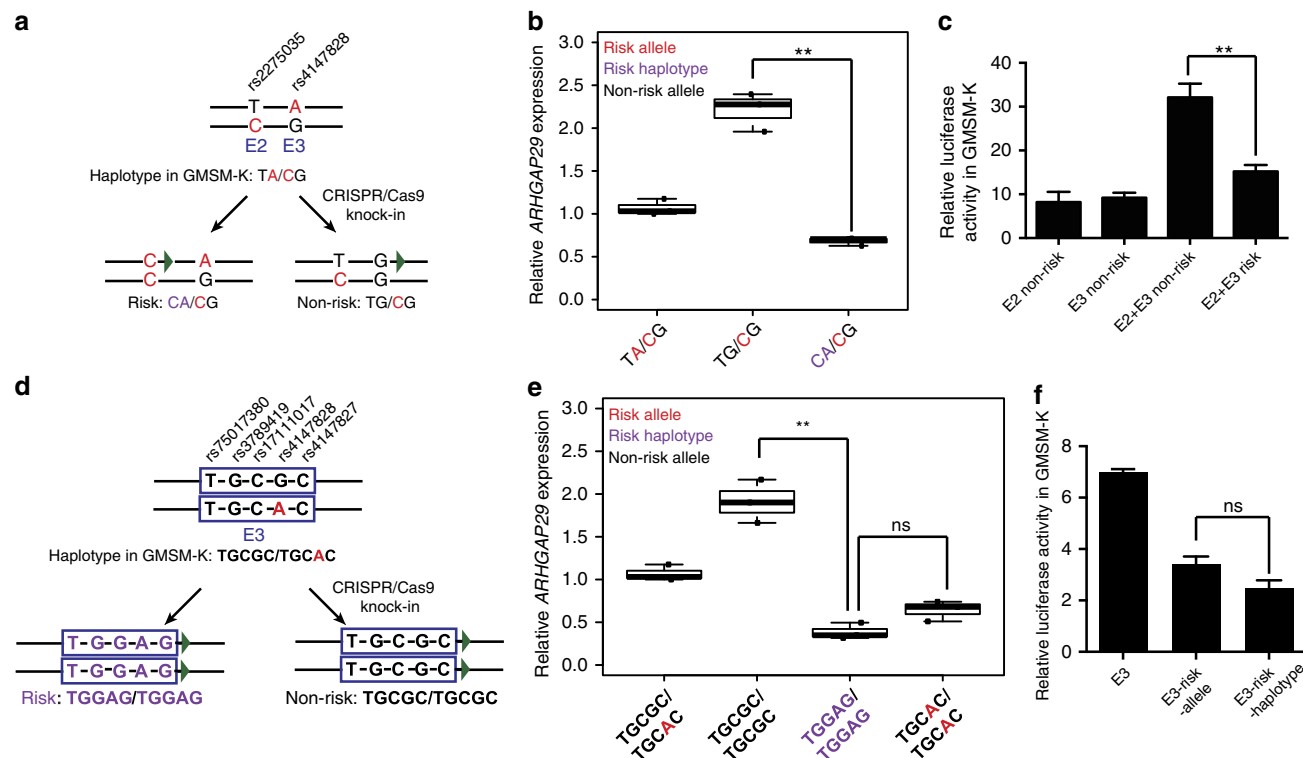

**Figure 6 | Risk haplotypes affect *ARHGAP29* expression.** (**a,d**) Schematic showing generation of oral epithelial cells of different haplotype. (**b**) Relative expression of *ARHGAP29* in oral epithelial cells with different haplotypes of rs2275035 and rs4147828; each dot represents mean expression of *ARHGAP29* in one isolated colony. $n = 3$ (colonies) for each group, *ARHGAP29* expression were measured for three times, and data represent means ± s.d., *t*-test, **$P < 0.01$. (**c**) Relative luciferase activity of fused E2 and E3 with risk or non-risk haplotype of rs2275035 and rs4147828 in GMSM-K oral epithelial cells. $n = 3$ for each group, luciferase activity was measured for three times for each group, data represent means ± s.d., *t*-test, **$P < 0.01$. (**e**) Relative *ARHGAP29* expression in oral epithelial cells with different haplotypes of rs75017380, rs3789419, rs17111017, rs4147828 and rs4147827. Note that TGCAC/TGCAC is the same as 'loxP-A/loxP-A' in Fig. 5h. $n = 3$ (colonies) for each group, *ARHGAP29* expression were measured for three times, and data represent means ± s.d., *t*-test, **$P < 0.01$. (**f**) Relative luciferase activity of E3 with different haplotype in GMSM-K oral epithelial cells. Haplotype of rs75017380, rs3789419, rs17111017, rs4147828 and rs4147827 in E3 is (TGCGC), E3-risk-allele (TGCAC), E3-risk-haplotype (TGGAG). $n = 3$ for each group, data represent means ± s.d., *t*-test, ns, non-significant.

improved by the risk allele relative to the non-risk allele. This protein is known to act as a transcriptional activator or repressor depending on context[42], and rare dominant-negative variants are present in Asian patients with NS CL/P (ref. 43). We carried out anti-KLF4 chromatin immunoprecipitation (ChIP) in lysates of GMSM-K cells, and found KLF4 binds E2 (Fig. 7a). Whereas the cell line is heterozygous for both risk and non-risk alleles at rs2275035, the ChIP-qPCR product was highly enriched for the risk-allele, showing KLF4 indeed binds this sequence more avidly (Fig. 7c). Given that E2 has lower enhancer activity when it harbours the risk allele at rs2275035 (Fig. 2a), we predicted that KLF4 binding inhibits the enhancer activity of E2. To test this, we transfected GMSM-K cells with the E2 luciferase reporter and an expression vector carrying human KLF4 cDNA. Luciferase levels

were lower in the context of KLF4 over-expression, and the reporter was more sensitive to such inhibition when it harboured the risk allele at rs2275035 (Fig. 7d). These findings suggest a mechanism by which the risk allele of rs2275035 could lead to lower enhancer activity of E2.

JASPAR predicted that rs4147828 (in E3) lies within a binding site of the transcription factor MAFB, and that the risk allele reduces the predicted quality of this site. MAFB promotes the differentiation of epidermis[44], and *MAFB* is at a locus that is strongly associated with risk for NS CL/P in the Asian population[3,22]. Using anti-MAFB qPCR-ChIP, we confirmed MAFB binds at rs4147828 in E3 (Fig. 7b), and sequencing of the ChIP-PCR product revealed a significant enrichment of the non-risk allele (Fig. 7c). In reporter assays using GMSM-K cells, forced

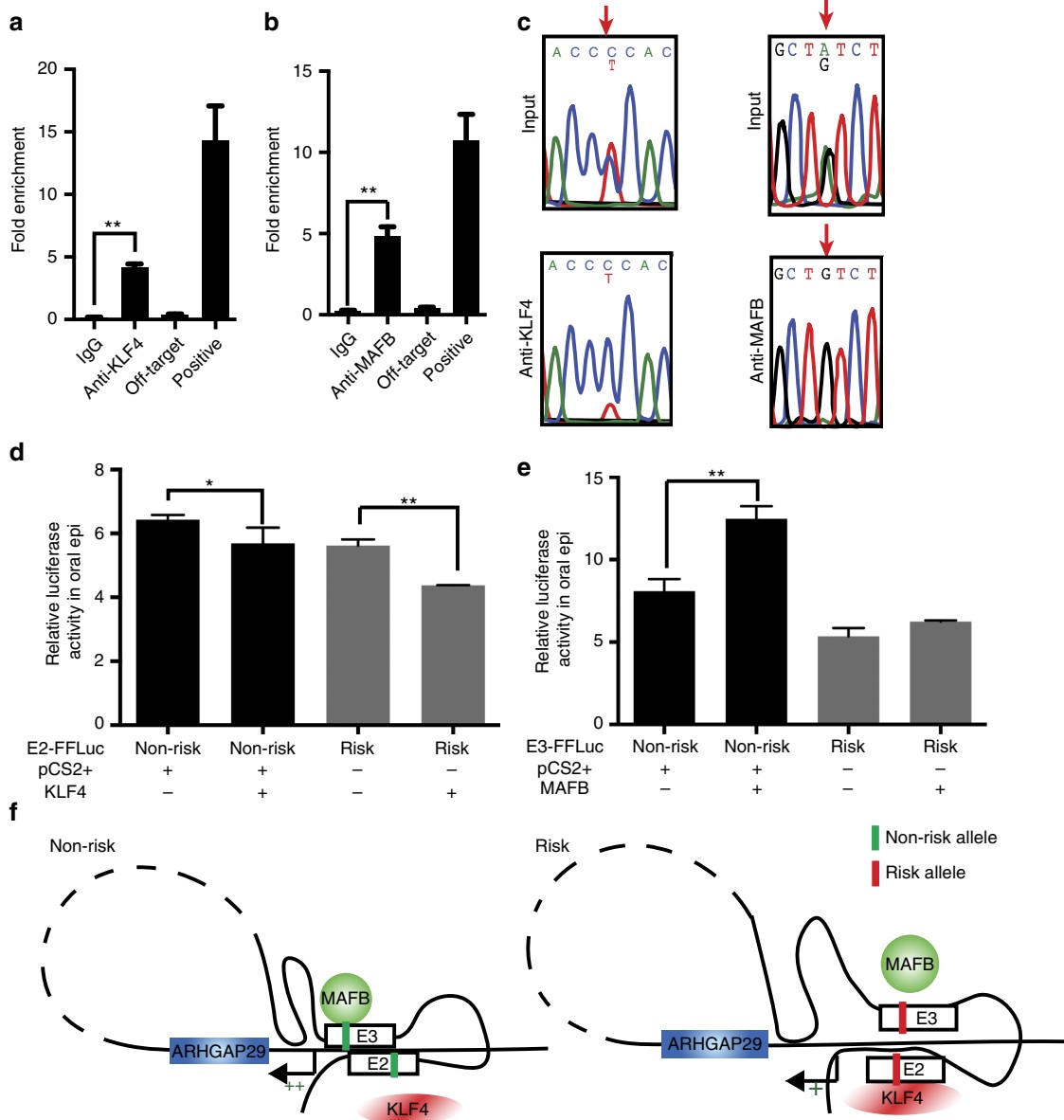

**Figure 7 | Different alleles of rs2275035 and rs4147828 affect bindings of NS CL/P-related transcription factors in *ARHGAP29* enhancers. (a,b)** ChIP-qPCR validating KLF4 binding around rs2275035 (**a**) and MAFB binding around rs4147828 (**b**). $n = 3$ for each group, data represent means ± s.d., t-test, **$P < 0.01$. (**c**) Sanger Sequencing of anti-KLF4 and anti-MAFB ChIP-PCR product. (**d**) Relative luciferase activity of E2 with different alleles of rs2275035 upon overexpression of KLF4. $n = 3$ for each group, luciferase activity was measured for three times for each group, data represent means ± s.d., t-test, **$P < 0.01$. (**e**) Relative luciferase activity of E3 with different alleles of rs4147828 upon overexpression of MAFB. $n = 3$ for each group, luciferase activity was measured for three times for each group, data represent means ± s.d., t-test, **$P < 0.01$. (**f**) Illustration of a possible model of *ARHGAP29* regulated by E2 and E3 with different genotypes.

expression of MAFB increased the enhancer activity of E3 when it contained the non-risk allele at rs4147828, but had no effect on an enhancer harbouring the risk-allele (Fig. 7e). These findings suggest that the risk allele at rs4147828 prevents binding of MAFB and inhibits ARHGAP29 expression, possibly by reducing the frequency with which E3 interacts with the promoter, which would be consistent with our 3C experiments (Fig. 4c). A working model is summarized in Fig. 7f.

## Discussion

We have demonstrated a systematic approach for investigating functional SNPs associated with NS CL/P, and it should be applicable to the investigation of other complex diseases (schematic of experimental pipeline shown in Supplementary Fig. 10). Starting with the ten SNPs in the 1p22 region that are most highly associated with NS CL/P in Asian case-parent trios, we conducted in vitro reporter assays in human oral epithelium and palate mesenchyme cells. Only nine of the candidate SNPs were found to reside within elements with enhancer activity, and only four were found to have allele-specific effects on enhancer activity. The utility of this assay, and that of the 3C-based method for identifying the best candidate as a directly causal gene, depends on having cell lines that are relevant to pathogenesis of the disorder in question. For instance, truly causal SNPs may be missed (false negative) if the disease-relevant enhancer in which it resides is inactive in the cell line model. Conversely, an inert SNP may be scored as causal (false positive) if by coincidence it has allele-dependent effects on an enhancer active in the cell line that is irrelevant to disease. Provided a relevant cell line available, tests of allele-dependence of enhancer activity in vitro are a potent and relatively scalable assay for use at loci with large numbers of risk-associated SNPs associated (for example, IRF6 (ref. 22)).

Reporter assays in zebrafish revealed the tissue specificity of enhancers containing the functional SNP candidates. This assay and analogous ones in mouse have the potential to greatly streamline the investigation of pathogenicity of any given SNP, particularly if the risk gene has a complex expression pattern. For instance in the present case, E2 and E3 enhancers were active in only a subset of the tissues in which mouse Arhgap29 (refs 45,46) and zebrafish arhgap29b (ref. 47) are expressed (for example, cranial epidermis, brain, oral mesenchyme and oral epithelium). Of note, expression was by far the strongest in cranial epidermis, and signalling from cranial epidermis might influence patterning of the underlying neural crest tissue[48]. The oral epithelium is also an excellent candidate tissue. Importantly, we can deduce that other domains of ARHGAP29 expression, for example, somites and endothelial cells, where E2 and E3 were inactive, are not involved in the pathogenicity of variation at 1p22. In principle, findings from such assays could strongly support one candidate as a causal gene for NS CL/P over another gene. However, in the present case it did not serve this purpose, as E2 and E3 were active in retina, where the ABCA4 gene is expressed. Also, E5, containing rs560426, exhibited low level enhancer activity in human oral epithelium cells in vitro, it did not do so, detectably, in zebrafish embryos, at least with the minimal promoters tested here.

CRISPR-Cas9-mediated HDR demonstrated that altering the risk allele at a single SNP, alone or in the context of a risk haplotype, could significantly alter the expression of ARHGAP29, simultaneously supporting the candidacy of functional SNPs and of the risk gene. This power of this assay again depends on having an appropriate cell line. Because of the poor colony-forming potential of the HEPM cell line, we did not manage to achieve HDR in them. Although induced pluripotent stem cells and embryonic stem cells have better colony-formation ability and are suitable for HDR (ref. 49), methods to convert such cells into oral epithelium and oral mesenchyme have not yet been reported.

Finally, ChIP revealed two SNPs affected the binding of specific transcription factors, MAFB and KLF4, potentially among others. Both proteins are members of gene regulatory networks active in epithelia, contribute to morphogenesis of the face (oral epithelium and cranial epidermis), and are associated with NS CL/P risk[3,21,28,43]. The evidence that the risk allele at a functional SNP affects MAFB binding, rs4147828, suggest there could be a gene-by-gene interaction between rs4147828 and the lead SNP at 20q12, where the MAFB gene resides. Detecting such interactions, unless they are unusually potent, requires very large sample sizes. We did not detect any statistical interaction between these loci in our recent large-scale GWAS (ref. 7). However, this required including 18 covariates in our regression model to account for population structure, limiting statistical power. This in no way limits the impact of identifying this biological interaction between ARHGAP29 and MAFB, two genes with limited research into their roles in craniofacial biology. The discovery of potential connections like these motivates further analyses of the gene regulatory networks controlling development of craniofacial tissues. Indeed despite mounting evidence that ARHGAP29 is the risk gene at 1p22, and that it participates in a network that includes KLF4, MAFB and IRF6 (refs 28,50), the role of ARHGAP29 during craniofacial development remains unknown.

## Methods

**SNP selection.** This study is based on association results from a targeted sequencing study of NS CL/P GWAS loci in case-parent trios from European and Asian ancestries[22]. The 1p22 locus was strongly associated with NS CL/P in Asian trios. There are several approaches for selecting SNPs for experimental pipelines, primarily based on LD with the lead SNP. Given that the 1p22 locus is within a region without strong LD patterns, we selected the top ten SNPs that had met all quality control metrics in our published study.

**Haplotype analysis.** Genotypes from the targeted sequencing were phased using BEAGLE (ref. 22). Phased haplotypes of 2–4 SNPs were analysed with a transmission disequilibrium test. Analyses were limited to case-parent trios from the Philippines, the subset of the Asian trios that drove the association signal. All analyses were implemented in R (v.3.3.0).

**Zebrafish maintenance.** D. rerio were maintained as described[51] in the University of Iowa Animal Care Facility (protocol no. 6011616). Zebrafish embryos were staged according to ref. 52 at 28.5 °C by hours or days post fertilization (h.p.f. or d.p.f.). And all the zebrafish experiments were performed in compliance with the ethical regulations in the Institutional Animal Care and Use Committee in the University of Iowa.

**Plasmid constructs.** All the candidate elements in 1p22 locus were cloned using the BAC RP11-109C4 as template and primers are documented in Supplementary Table 9. Products were cloned into pENTR/D-TOPO plasmid (Life Technologies, Carlsbad, CA, USA) for Sanger Sequencing validation. After comparing the genotypes of the cloned elements with the risk or non-risk genotype for each SNPs reported in our previous study[22], site-directed mutagenesis was employed to get the elements with non-risk or risk allele. After sequencing confirmation, candidate elements were shuttled into cFos-FFLuc plasmid for in vitro luciferase assay, and cFos-GFP (ref. 53) or ZED (ref. 54) plasmids for zebrafish in vivo enhancer assay.

Homology arms used for HDR-mediated by CRISPR-Cas9 were cloned using the elements verified in the pENTR/D-TOPO plasmids mentioned above as templates, then digested and inserted to the upstream and downstream of loxP-neoR-loxP cassette of pEasy-Flox (a gift from Klaus Rajewsky (Addgene plasmid #11725)).

KLF4 cDNA (RefSeq: NM_004235.4) was obtained from Stratagene. The coding sequence of human MAFB was cloned directly from human genomic DNA. The coding sequence of human KLF4 and MAFB was cloned into pENTR/D-TOPO plasmid for Sanger sequencing validation, after which they were shuttled into pCS2 + -destination plasmid for over-expression assay.

**Cell culture and transfections.** GMSM-K human embryonic oral epithelial cell line (a kind gift from Dr Daniel Grenier)[30] was maintained in keratinocyte serum-free medium (Life Technologies) supplemented with EGF and bovine pituitary extract (Life Technologies). All cells were incubated at 37 °C in 5% $CO_2$. Human

embryonic palatal mesenchyme cells (HEPM)[31] were purchased from ATCC (ATCC CRL-1486) and maintained in ATCC-formulated Eagle's minimum essential medium (ATCC) supplemented with 10% fetal bovine serum (Life Technologies) and 1% antibiotic-antimycotic (Life Technologies). The 293FT cell line (human embryonic kidney cell line) was purchased from Invitrogen and maintained in DMEM supplemented with 10% fetal bovine serum (Life Technologies) and 1% antibiotic-antimycotic (Life Technologies). Early passage of 293FT cells were used for lentivirus package. All the cell lines used in this study were tested to be free of mycoplasma contamination and authenticated by genetic profiling using polymorphic short tandem repeats.

To generate lentivirus coding Cre recombinase, lentivector pLOX-CW-CRE (ref. 55) was co-transfected with pMD2.G and psPAX2 (kind gift from Dr Didier Trono) into 293FT cells[56] using Lipofectamine 3000 (Life Technologies). Lentivirus was collected and purified everyday from second day to fourth day after transfection.

**Dual luciferase activity assay.** For dual luciferase activity assay, each reporter construct was co-transfected with Renila Luciferase plasmid for three biological replicates. Briefly, plasmids were electroporated into GMSM-K cells ($1 \times 10^6$ per transfection) with Amaxa Cell Line Nucleofector Kit V (Lonza, Cologne, Germany) using Nucleofector II (Lonza) (program: X-005), and plasmids were electroporated into HEPM cells ($1 \times 10^6$ per transfection) with Amaxa Basic Nucleofector Kit for Primary Mammalian Fibroblasts (Lonza) using Nucleofector II (Lonza) (program: U-020). The Dual-Luciferase Reporter Assay System (Promega, Madison, WI, USA) and 20/20n Luminometer (Turner Biosystems, Sunnyvale, CA, USA) were employed to evaluate the luciferase activity when cells were approximately 95% confluent at 72 h post-transfection following the manufacturer's instructions. Relative luciferase activities were calculated by the ratio between the value for firefly and Renilla luciferase activities. Three measurements were made for the lysate from each transfection group. All quantified results are presented as mean ± s.d. Student t-test was used to determine statistical significance.

**Zebrafish enhancer in vivo reporter assay.** For transient transgenic assay, the cFos-GFP vector with each candidate regulatory element was injected along with the tol2 mRNA (20–30 ng μl$^{-1}$) into at least 200 one-cell stage embryos. The embryos were screened at 6 h.p.f., 24 h.p.f., 48 h.p.f. and 4 d.p.f. for documenting enhancer activity in different embryo development stages. A consistent GFP expression pattern in a minimum of 10% of injected fish was the criterion for tissue-specific enhancer activity (recorded in Supplementary Table 3), which will be sufficient to predict their pattern in F1. For E2 and E3 that exhibit consistent craniofacial enhancer activity, we shuttled them into ZED vector to establish stable transgenic line. F0 adults of E2 and E3 were outcrossed with NHGRI-1 line[57], F1 with RFP (transgenic marker) from three F0 founders of each line were maintained. F2 embryos were documented for detailed enhancer pattern at 6 h.p.f., 24 h.p.f., 30 h.p.f. and 4 d.p.f. Representative enhancer patterns in each stage were photographed in bright field, epi-fluorescent illumination on a Leica DMRA2 compound microscope with a colour 12 bit 'QIClick' camera (Qimaging). Embryos at 4 d.p.f. were fixed and cryo-sectioned for photograph. Live embryos at 4 d.p.f. were also photographed in Zeiss 700 confocal microscope with a × 20 per 0.5 NA objective lens (Carl Zeiss).

To compare the difference between E2 and E3 with risk or non-risk allele in vivo, cfos-EGFP vector with E2 or E3 with risk or non-risk allele were injected into more than 200 embryos respectively for three times. Twenty-four hours after injection, unhealthy or dead embryos were excluded from this study. Every healthy embryo was examined under the fluorescent microscope to compare the tissue-specific GFP pattern in an unblended way. The investigator was not blinded to group allocation during experiment. The results are recorded in Supplementary Table 4.

**Real-time PCR analysis.** To compare gene expression of cells with different genotype, same amount of cells were plated in six-well plate When cells reached 80% confluence, total RNA were isolated using RNeasy mini kit (Qiagen, Hilden, Germany) and treated with RNease-free DNaseI (Promega, Madison, WI, USA). Reverse transcription was performed by High Capacity cDNA Reverse Transcription Kits (Applied BIosystems, Foster city, CA, USA). Real-time PCR was performed in CFX96 TouchReal-Time PCR Detection System using TaqMan-probe based gene expression assay (Life Technologies) (or SYBR-Green qPCR (Bio-rad) (Supplementary Table 11). For SYBR-Green qPCR, primers were synthesized by IDT (Coralville, IA, USA). GAPDH was used as internal control.

**Chromatin conformation capture and genotype of 3C product.** Primers for 3C-qPCR were designed using Primer3 Plus online software (http://www.bioinformatics.nl/cgi-bin/primer3plus/primer3plus.cgi) and tested against EcoRI-digested, ligated BAC templates library (RP11-109C4 and RP11-979G24 for promoter region of ARHGAP29, RP11-937N17 for promoter of ABCD3, RP11-826E4 for the region in the neighbouring TAD, listed in Supplementary Table 8) by serial dilution and melt-curve analysis to ensure specific and linear amplification. Also, such libraries were used for quantification of all the 3C PCR product. The 3C-qPCR assay was performed according to standard protocol[58] for at least two

replicates. Briefly, about $1 \times 10^7$ GMSM-K cells were crosslinked with 1% formaldehyde at room temperature for 10 min, followed by glycine quenching and cell lysis. Nuclei were resuspended in 500 μl Buffer EcoRI, then lysed with 0.2% SDS, followed by SDS sequestration with 1.2% Triton X-100. Lysates were digested overnight at 37 °C with EcoRI (New England BioLabs, Ipswich, MA, USA), and the restriction enzyme then inactivated for 1 h at 65 °C with 1.6% SDS. Ligation was performed with T4 ligase in $1.15 \times$ T4 Ligase buffer at 16 °C for 4 h. Finally, the ligation products were purified with phenol-extraction protocol. 3C ligation products were quantified with three replicates by SYBR Green qPCR. The 3C signals were quantified using the standard curve obtained from BAC library, then further normalized to an undigested GAPDH region for input loading control. Digestion efficiencies were determined with SYBR Green qPCR by comparison of aliquots taken pre- and post-digestion using primer pairs that amplified a region in each interrogated fragment spanning an EcoRI digestion site. Since the EcoRI digestion sites are less than 500 bp away from transcription start sites of ARHGAP29 and ABCD3, we designed two primers anchoring to both ends of EcoRI digested fragment flanking ARHGAP29 and ABCD3 transcription start sites, and compared the relative interaction frequency. Since the results from both sets are similar, we only present the interaction between 1p22 region to the EcoRI in the upstream to ARHGAP29 and ABCD3 transcription start sites (using primers 'Detection of EcoRI fragment interacting ARHGAP29 (NM_001328667.1) promoter_1' and 'Detection of EcoRI fragment interacting ABCD3(NM_002858) promoter_1' in Supplementary Table 10).

To genotype the 3C product, we did PCR using one primer anchoring to the promoter of ARHGAP29, and one primer for ChIP-qPCR for rs4147828 or rs2275035. Raw PCR product was gel-extracted for Sanger sequencing.

**Chromatin immunoprecipitation quantitative PCR.** Chromatin immuno-precipitation quantitative PCR (ChIP-qPCR) was performed according to standard protocol[59]. Briefly, about $1 \times 10^7$ GMSM-K cells were chemically cross-linked by formaldehyde in room temperature for 10 min followed by quench using glycine. Cells were resuspended in cell lysis buffer (150 mM NaCl, 10 mM HEPES, pH 7.4, 1.5 mM $MgCl_2$, 10 mM KCl, 0.5% NP-40, 0.5 mM DTT) and diluted with dilution buffer (150 mM NaCl, 16.7 mM Tris, pH 7.5, 3.3 mM EDTA, 1% Triton X-100, 0.1% SDS, 0.5% Na-Doc) to obtain an aliquot of $1 \times 10^6$ cells for ChIP. Cell lysates were sonicated by Focused-Ultrasonicator (Covaris) to shear the cross-linked DNA. Sheared DNA was incubated with Dynabeads (Life Technologies) pretreated by KLF4 antibody (Abcam, ab72543, lot#: GR179695-1, 5 μg per 100 μl Dynabeads) or MAFB antibody (P-20, SantaCruz, sc-10022, lot#: L2115; 5 μg per 100 μl Dynabeads) or IgG Dynabeads (Life Technologies) as control overnight. The next day beads were washed sequentially with low salt wash buffer, high salt wash buffer, LiCl wash buffer and TE buffer. Cross-linked DNA bound to the beads was eluted using SDS solution, and cross-linking reaction was reversed by overnight incubation with 5 M NaCl. DNA was purified and precipitated by phenol:chloroform and EtOH. Purified DNA was subjected to qPCR using primer listed in Supplementary Table 12. We also used two previously reported ChIP-seq peaks of KLF4 (ref. 60) (chr1:94,510,876-94,511,769(hg19)) and MAFB (ref. 44) (chr1:94,703,328-94,703,330 (hg19)) in NHEK as positive controls, and one off-target (negative control chr20:38,034,907-38,034,910(hg19)).

**CRISPR-Cas9-mediated knockout and knockin.** All the gRNAs used in this study were designed by CRISPRscan (http://www.crisprscan.org/), and cloned into px330 (a gift from Feng Zhang (Addgene plasmid #42230)). Sequences of all the gRNAs target sequences are documented in Supplementary Tables 13 and 14.

Before genome editing in GMSM-K cells, genotyping of the three SNPs and phasing of the two haplotypes mentioned above were performed. Briefly, a set of primer flanking rs2275035 and rs4147828 (reverse primer for ChIP-qPCR targeting rs2275035 and reverse primer for ChIP-qPCR targeting rs4147828) were used to clone the genome DNA sequence of GMSM-K cells with StrataClone Blunt PCR Cloning Kit (Agilent Technologies, Santa Clara, CA, USA). Sanger sequencing of ten resulting plasmids identified the phase of the haplotypes.

There is a high-scoring gRNA target site just 17 bp away from 5′ of rs560426 (in E5). We co-transfected cells with a single stranded donor template and a plasmid encoding the relevant gRNA and Cas9 (that is, the gRNA and CAS9 expression plasmid). We applied selection, isolated single cells and grew colonies from them, and used polyacrylamide gel (PAGE)-gel-based genotyping[61] or high-resolution melting analysis[62,63] followed by Sanger sequencing to identify colonies that were homozygous for this 10-bp deletion, the risk allele or the non-risk allele (Supplementary Fig. 4).

To knockout E2 and E3, we cotransfected GMSM-K with two CRISPR-Cas9 plasmids expressing two gRNAs flanking target enhancer regions with Amaxa Cell Line Nucleofector Kit V (Lonza) using Nucleofector II (Lonza) (program: T-002) along with empty pFlox-Easy plasmid for neomycin transient selection. Forty-eight hours post-transfection, cells were treated with G418 (Sigma-Aldrich, St Louis, MO, USA) (1200 ng ml$^{-1}$) for 3 days, and serial dilution in 96-well plates were used to get single cell colony. Cell colonies were 'picked-up' by pipette tip as template for PCR screening. Screening processes are described in Supplementary Fig. 6.

The HDR in rs560426 was performed using single-strand DNA as donor template. Ultramers with 50 bp flanking desired allele or 10-bp deletion near

rs560426 were synthesized by IDT. Five microlitres single strand DNA (10 μM) with PX330 expressing gRNA targeting rs560426 were co-transfected into GMSM-K by electroporation as described above. Forty-eight hours after transfection, serial dilution approach was used to get single cell clone. When single cell colonies were formed, colony was screened by PAGE to obtain homozygous rs560426 or the colonies with homozygous deletion. If the genotype of rs560426 in the colonies screened was heterozygous, at least three bands will be seen in PAGE, but PCR product from colonies with homozygous rs560426 will have only one band. After screening, genotypes of the homozygous colonies were confirmed using Sanger Sequencing.

The HDR for the other two SNPs and haplotypes were performed using plasmids as donor templates. Donor plasmid with different genotypes or haplotypes were cotransfected with PX330 expressing the relevant gRNA. The nucleofected GMSM-K cells were plated in 10 cm dishes, and 48 h after transfection, were treated with G418 (Sigma-Aldrich) (800 ng ml$^{-1}$) and Ganciclovir (InvivoGen, San Diego, CA, USA) (4 μM) for 10 days. After drug selection, colonies were manually picked from each dish, screened and expanded. We avoided picking colonies growing close to each other. For screening the colonies with neomycin cassette in both alleles, we employed high-resolution-melting-curve-assay (CFX-Connect, Bio-rad, Hercules, CA, USA) with primers flanking the cassette. Sanger sequencing was employed to confirm the genotype. In order to delete the neomycin cassette, we delivered lentivirus expressing Cre recombinase. After infection, colonies were screened using PCR. Sanger sequencing was used to confirm the genotype.

**In vitro wound healing assay.** For each genotype, same number of cells from three colonies of GMSM-K cells were plated independently in six-well plate. Scratch assay were performed according to standard protocol[64]. Briefly, scratches were generated with a P200 tip, and static images of three fixed spots along each scratch on 0 h (right after scratch), 3, 6, 12, 18, and 24 h after scratching. Static images were recorded on a Leica DMRA2 compound microscope with a colour 12 bit 'QIClick' camera (Qimaging). Intervals of each spot were measured for three times with ImageJ (https://imagej.nih.gov/ij/). Since the scratches in some wells almost closed at 18 h, we only included the distance of cell migration within 18 h after scratch was made. Distances were presented as mean ± s.d. Representative photos are presented in Supplementary Fig. 8.

**Statistical analysis.** For luciferase assay and RT-PCR, each group includes three biological replicates, and for each biological replicate, three technical replicates were measured to ensure the statistical power. For CRISPR-engineered cells, at least three independent colonies were picked up for experiment (detailed numbers are recorded in Supplementary Fig. 6). Statistical analyses were carried out using two-tailed, unpaired Student's t-test. Before t-test, normal distribution of all the data were checked using normality test and equality of variances were checked using F-test implemented in R (v.3.2.5).

**Data availability.** The data that support the findings of this study are available from the corresponding author on reasonable request.

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

## Acknowledgements

We are grateful for excellent technical assistance and animal husbandry from Greg Bonde. We are grateful to Johann Eberhart from the University of Texas at Austin for helping us interpret zebrafish craniofacial anatomy. We thank Martine Dunnwald for discussion of the *in vitro* scratch assay, Adam Dupuy for discussion of CRISPR experiment design, and Jeffery Murray, supported by DE08559, for permitting analysis of unpublished targeted resequencing results. We thank Axel Visel and Cailyn Spurrell for preliminary analyses of enhancer activity in mouse embryos of a subset of elements discussed here. Research reported in this publication was supported by the National Institute of Health grants DE023575 (R.A.C.), DE025060 (E.J.L.), DE016148 (M.L.M.), HG005925 (M.L.M.), DE014581 and DE018993 (T.H.B.), and by the National Natural Science Foundation of China 81400477 (H.L.). The content is solely the responsibility of the authors and does not necessarily represent the official views of the National Institutes of Health.

## Author contributions

H.L. planned experiments, performed the cell-based and fish-based experiments and prepared the manuscript. E.J.L., J.C.C. and M.L.M. performed analysis of human genetic data. T.H.B. contributed to GWAS and resequencing experiments that identified the risk-associated SNPs at 1p22. A.C.L. planned experiments. R.A.C. planned experiments, supervised the project and prepared the manuscript.

## Additional information

**Competing financial interests:** The authors declare no competing financial interests.

**Publisher's note**: 

