## [Peer Review File · Nature Communications]

Reviewers' comments:

Reviewer #1 (Remarks to the Author):

Nonsyndromic cleft lip with or without cleft palate (NSCL/P) is a rather frequent human malformation, with complex etiology, which is disturbing and stressful to patients and their families. In the last years, numerous GWAS performed in different (mainly European and Asian) ethnicities have led to the identification of about 20 genetic risk loci, virtually all of which are located in non-coding regions of the human genome. This observation, which is common in complex trait genetics and has been observed for many different phenotypes, leads to the hypothesis that the risk variants are involved in regulatory mechanisms. However, the functional follow-up of this hypothesis is depending of (i) the availability of relevant tissue, (ii) the knowledge about the actual causal SNPs among all those significant in a GWAS, and (iii) the availability of adequate and sensitive experimental assays. In NSCL/P, at least one of these points have not (yet) been available or systematically applied. In the present study, the authors now present a strategy that integrates genetic data, reporter assays, genome editing, cellular and fish models as well as transcription factor binding data in order to comprehensively follow-up one of the first risk loci identified for NSCL/P, i.e., the 1p22 locus. Among a list of candidate variants and candidate enhancers they identify three that show conclusive activity in relevant cell types and tissues as well as allele-specific effects, suggesting them as likely pathogenic variants. Finally, additional experiments lead to the presentation of a novel molecular framework by which the 1p22 risk region contributes to NSCL/P disease.

Points to be considered:

1) The manuscript is sometimes hard to read, in particular the Results section. There is a lot of experimental details presented (which should be moved to the Methods section), and also a lot of repetitions e.g. of SNP-IDS. In particular, the authors often describe their results enhancer by enhancer, making the content very redundant. The authors should consider providing the results in more structured ways (for instance, Tables) and only summarize the respective findings in text-style in the Results section. Also, at many points the authors claim that something was significant, however, no P-values were provided (in most cases, P-values were found in the Figures though). P-values should be added to the text for convenience of the reader.

2) According to their results, three variants (and two haplotypes) remain as those that are likely pathogenic. The selection of their SNPs, however, was based on Asian data available from a previous resequencing study. If variants are truly causative, I would hypothesize that these variants / haplotypes should also be associated in other ethnicities such as Europeans. As the authors have resequencing data from Europeans as well (see Leslie et al. 2015), association data should be checked for these SNPs in these populations.

3) Experimental details: In their first selection of variants to be included in the analysis, the authors select the top 10 of associated SNPs from a previously published resequencing study located in regions with positive enhancer activity. As negative control, regions without any enhancer-mark were taken. In my opinion, this might reflect a bias already in study design, making the chance to identify the associated SNPs as active high a priori. Why weren't elements included that either had associated risk SNPs without enhancer activity, or active enhancers without SNPs included? This would have allowed for a more comprehensive understanding of the genetic risk / enhancer architecture at 1p22.

Minor points:

- In the Abstract the authors introduce the phenotype as OFC (orofacial clefting) while it is NSCL/P throughout the remainder of the text (which is actually more accurate). The authors should replace OFC by NSCL/P in the Introduction. In that same line, the authors sometimes write NS CL/P or NSCLP. Please check for consistency.

- Introduction, line 62: "... GWAS... successful in identifying several potentially causal genes for NSCL/P". I do not think this is true – GWAS have indicated genetic risk loci in general, and for some of them, resequencing or additional experiments have provided some evidence for candidate genes. In the next sentence, the authors state that "by its nature, [GWAS] cannot distinguish between truly causal variants". This was likely to be true for normal GWAS using genotyped variants only, however, with the advance of imputation methods and statistical approaches such as credible SNP analyses, this gets more and more possible. This being said I still agree with the authors that (1) these approaches have not yet been conclusively applied to NSCL/P, and (2) overall, a functional effect would still need to be demonstrated as, again, this would all be statistics...

- Introduction: The authors summarize the support for ARHGAP29 to be the candidate gene at 1p22 by adding evidence from a resequencing study in which "coding variants in ARHGAP29 are strongly associated with CL/P". In the cited study by Leslie et al., some rare variants were found, however, they mostly had reduced penetrance. Therefore I would suggest to omit the word "strong".

- The authors applied genome editing to their cell systems, however, they also mentioned that the genomic context might be important, and therefore moved to the zebrafish for the reporter assays. Why was genome editing not applied to zebrafish as well?

- Measurements were often taken at certain timepoints, for instance, Luciferase activity in cells was monitored after 72h while the cell scratch assay was measured after 18h. This seems arbitrary as no references / reasons therefore were provided.

- In their firefly / renilla assays, both luciferases were encoded by different plasmids, representing a technical bias that was not corrected for (nor adequately addressed). For instance, more of fewer plasmids of one type could have been introduced in one cell while a different ratio would have been applied to another one. Why didn't the authors use plasmids that contain both luciferases?

- The authors used molecular engineering techniques to generate the allele combinations that they wanted to investigate. Why were elements not cloned from patient DNA for which these genotypes were readily available?

- The authors state that the allele-specific effect was not detectable by eye in the zebrafish experiments, probably due to a low effect size. I agree, but why didn't the authors aim at quantifying it?

- Results, line 246. The authors state that "risk allele at rs4147828 disrupts assembly of the protein complex mediating interactions between E3...". This is an overstatement or point for discussion, because the results provided by the authors only show the extent of interaction, but no evidence at all at molecular level.

- The ID of the craniofacial element used from Attanasio et al should be provided in the text.

- Had rare variants been observed in the E2, E3 and E5 regions in the resequencing study?

- Discussion, line 424: "... the assays are subject to false negatives (although false-positives are not probable)". The authors should explain or provide references herefore. Couldn't a regulatory element be highly craniofacial-specific and active only during exact relevant timepoints and, hence, be missed?

Reviewer #3 (Remarks to the Author):

This is, in my opinion, a very thorough and detailed investigation into the genetic mechanisms contributing to NS CL/P. I was particularly impressed at the complementary lines of evidence used to clearly link regulatory variants with enhancer activity in relevant tissues to target genes and ultimately to a cellular phenotype relevant to NS CL/P. The manuscript is clearly written, and the findings well-justified. I further agree with the authors that this is an exemplary study for others investigating the potential regulatory mechanisms underlying genetic associations.

My only and minor concern is that it would be helpful to replicate the structure of the various assays in the figure legends rather than having to find them in the supplementary methods.

Again, a well done study of which I am highly enthusiastic.

Sincerely,
Timothy E. Reddy
Duke University

Reviewer #4 (Remarks to the Author):

In the manuscript "Identification of common, non-coding variants at 1p22 that are pathogenic for non-syndromic cleft lip with or without cleft palate", Liu, et al elegantly used a multipronged approach for identifying the potential mechanism of pathogenesis of three risk-associated SNPs. Overall, the experiments are very well-designed and the results are convincing because all the necessary control groups and replicates are included. In my opinion, the only missing information in this study is the connection between the reduction of ARHGAP29 expression and the cell migration defects, since this seems to be what the authors imply. If so, a simple shRNA knockdown experiment should be provided to clarify the role of ARHGAP29.

Response to Referees Letter

Reviewer 1

1) The manuscript is sometimes hard to read, in particular the Results section. There is a lot of experimental details presented (which should be moved to the Methods section), and also a lot of repetitions e.g. of SNP-IDS. In particular, the authors often describe their results enhancer by enhancer, making the content very redundant. The authors should consider providing the results in more structured ways (for instance, Tables) and only summarize the respective findings in text-style in the Results section. Also, at many points the authors claim that something was significant, however, no P-values were provided (in most cases, P-values were found in the Figures though). P-values should be added to the text for convenience of the reader.

We agree with and have adopted the reviewer's suggestions for ways to make the paper more readable (changes marked with a vertical line in the left hand margin). The most significant changes were that we shortened the Results section on the outcome of in vitro enhancer assays, and moved details of the genome editing to the Methods section. We constructed a table summarizing the in vitro reporter assays, but it is so large that it did not fit well in the main paper and we therefore put in the supplement. We now include p-values in the text. We also moved the cartoon model from the supplement into Figure 7. We hope that these changes in the writing and the figures have made our story easier to follow.

2) According to their results, three variants (and two haplotypes) remain as those that are likely pathogenic. The selection of their SNPs, however, was based on Asian data available from a previous resequencing study. If variants are truly causative, I would hypothesize that these variants / haplotypes should also be associated in other ethnicities such as Europeans. As the authors have resequencing data from Europeans as well (see Leslie et al. 2015), association data should be checked for these SNPs in these populations.

The reviewer is correct that we would hypothesize causative variants to have the same functional effect in all populations. However, this does not mean that an association will be observed in all populations. The ability to detect an association depends on the minor allele frequency of the SNP, the effect of the SNP (odds ratio), and the sample sizes in the study. Only 300 European trios were resequenced and the power to detect an association in this sample was limited except for loci with very strong effects. The 1p22 locus has been independently replicated in other populations including Europeans and admixed Latino populations (for example Cura et al 2016, Birth Defects Research; Leslie et al. 2016, Hum. Molecular Genetics).

3) Experimental details: In their first selection of variants to be included in the analysis, the authors select the top 10 of associated SNPs from a previously published resequencing study

located in regions with positive enhancer activity. As negative control, regions without any enhancer-mark were taken. In my opinion, this might reflect a bias already in study design, making the chance to identify the associated SNPs as active high a priori. Why weren't elements included that either had associated risk SNPs without enhancer activity, or active enhancers without SNPs included? This would have allowed for a more comprehensive understanding of the genetic risk / enhancer architecture at 1p22.

The reviewer is correct that we deliberately selected chromatin elements for which previous evidence suggested enhancer activity, and thus it is not surprising that these elements were more active than control regions. We indeed performed enhancer tests on elements that harbor risk-associated SNPs but lack chromatin marks indicative of enhancer activity (elements E4 and E5, see Fig 1). We did not do the converse because, although elements lacking risk-associated SNPs may have enhancer activity, testing them would not advance our effort to recognize functional SNPs. Note that we excluded an SNP that does not reside within an element that exhibits enhancer activity. It is possible that this was a false negative because, as mentioned in the Discussion, causal SNPs will be missed by our assays if our cell lines are inappropriate. However, the observation that some but not all SNPs had allele-specific effects on enhancer activity in these cell lines suggests that they were appropriate.

Minor points:

- In the Abstract the authors introduce the phenotype as OFC (orofacial clefting) while it is NSCL/P throughout the remainder of the text (which is actually more accurate). The authors should replace OFC by NSCL/P in the Introduction. In that same line, the authors sometimes write NS CL/P or NSCLP. Please check for consistency.

In the revised manuscript, we use NS CL/P throughout, except in the title where we use "orofacial clefting" because of the title must 15 words or less.

- Introduction, line 62: "... GWAS... successful in identifying several potentially causal genes for NSCL/P". I do not think this is true – GWAS have indicated genetic risk loci in general, and for some of them, resequencing or additional experiments have provided some evidence for candidate genes.

We agree with the reviewer and in the revised manuscript, we the offending sentence has been replaced with the following one: "The GWAS approach has been unusually successful in identifying loci in which variation contributes significantly to risk for NS CL/P, in comparison to its degree of success for other complex diseases."

In the next sentence, the authors state that "by its nature, [GWAS] cannot distinguish between truly causal variants". This was likely to be true for normal GWAS using genotyped variants only, however, with the advance of imputation methods and statistical approaches such as credible SNP analyses, this gets more and more possible. This being said I still agree with the authors that (1) these approaches have not yet been conclusively applied to NSCL/P, and (2) overall, a functional effect would still need to be demonstrated as, again, this would all be statistics...

We are in agreement with the reviewer. When SNPs are in strong LD, even very large studies

of patients and controls will never be able to distinguish between causal SNPs and rider SNPs.

- Introduction: The authors summarize the support for ARHGAP29 to be the candidate gene at 1p22 by adding evidence from a resequencing study in which “coding variants in ARHGAP29 are strongly associated with CL/P”. In the cited study by Leslie et al., some rare variants were found, however, they mostly had reduced penetrance. Therefore I would suggest to omit the word “strong”.

We appreciate that the reviewer caught this mistake and have adopted the suggested change in wording.

- The authors applied genome editing to their cell systems, however, they also mentioned that the genomic context might be important, and therefore moved to the zebrafish for the reporter assays. Why was genome editing not applied to zebrafish as well?

Because of significant divergence in enhancer sequences between humans and zebrafish, there is no way to identify the nucleotide in the zebrafish genome that is analogous to a given risk-associated SNP.

- Measurements were often taken at certain timepoints, for instance, Luciferase activity in cells was monitored after 72h while the cell scratch assay was measured after 18h. This seems arbitrary as no references / reasons therefore were provided.

In the vast literature of luciferase reporter assays, luciferase is commonly monitored anywhere 24-96 hours after transfection. In the revised manuscript in the Methods we clarify that we transfected cells at 50% confluency and monitored luciferase when they reached 95% confluency, and that this occurred at 72 hours. Importantly, given that all conclusions from the in vitro reporter assays came from comparisons, the conclusions are not particularly sensitive to the time when luciferase is monitored. We chose 18 hours for photographing the migrating cells because this the time when the most rapidly migrating cells had reached the scratch, closing the wound.

- In their firefly / renilla assays, both luciferases were encoded by different plasmids, representing a technical bias that was not corrected for (nor adequately addressed). For instance, more of fewer plasmids of one type could have been introduced in one cell while a different ratio would have been applied to another one. Why didn't the authors use plasmids that contain both luciferases?

We agree that a single plasmid with two cassettes would be preferable, however it is utterly routine in the field to use two plasmids. More importantly, other results in the paper, most convincingly the genome editing, support the conclusions drawn from the in vitro reporter assays.

- The authors used molecular engineering techniques to generate the allele combinations that they wanted to investigate. Why were elements not cloned from patient DNA for which these genotypes were readily available?

Engineering the various allele combinations within elements ensured that we knew the

differences between the elements. It is true that an alternative approach would have been to amplify them from patient DNA, but this would not necessarily have been easier.

- The authors state that the allele-specific effect was not detectable by eye in the zebrafish experiments, probably due to a low effect size. I agree, but why didn't the authors aim at quantifying it?

The level and the spatial distribution of reporter expression in transgenic zebrafish are subject to position-of-integration effects. By comparing several independent isolates, as we did, it is possible to gain some confidence in the true spatial expression domain. However, to compare quantitative differences, it is essential to use single-integration site transgenics, i.e., using the PhiC31 system. Although there are some promising early papers on the application of this system in zebrafish, bugs still need to be worked out. Also, the system is not widely available.

- Results, line 246. The authors state that "risk allele at rs4147828 disrupts assembly of the protein complex mediating interactions between E3...". This is an overstatement or point for discussion, because the results provided by the authors only show the extent of interaction, but no evidence at all at molecular level.

We have altered the indicated sentence to read, "This result suggests the risk allele at rs4147828 disrupts the interactions between enhancer E3 and the ARHGAP29 promoter."

- The ID of the craniofacial element used from Attanasio et al should be provided in the text.

The element is now identified as mouse element 435.

- Had rare variants been observed in the E2, E3 and E5 regions in the resequencing study?

Yes, rare and low-frequency variants were observed in E2, E3, and E5. However, our previous analyses of rare variants did not identify an over-transmission of these variants to affected offspring.

- Discussion, line 424: "... the assays are subject to false negatives (although false-positives are not probable)". The authors should explain or provide references herefore. Couldn't a regulatory element be highly craniofacial-specific and active only during exact relevant timepoints and, hence, be missed?

We have tried to clarify this point by changing the relevant sentences in the Discussion to the following: "For instance, a truly causal SNPs may be missed (false negative) if the disease-relevant enhancer in which it resides is inactive in the cell line tested. Conversely, an inert SNP may be scored as causal (false positive) if by coincidence it has allele-dependent effects on an enhancer that is active in the cell line but irrelevant to disease."

Reviewer #3 (Remarks to the Author):

"..It would be helpful the replicate structure of the various assays in the figure legends rather than having to find the in the supp methods."

In several cases (changes indicated by vertical lines in the left hand margin) we have altered the figure legends to make the nature of the assays more explicit.

Reviewer #4 (Remarks to the Author):

“..In my opinion, the only missing information in this study is the connection between the reduction of ARHGAP29 expression and the cell migration defects, since this seems to be what the authors imply. If so, a simple shRNA knockdown experiment should be provided to clarify the role of ARHGAP29.”

We carried out and recently published the requested shRNA experiment. The revised text now reads, “ARHGAP29 inactivates RhoA³⁷ and RhoA regulates keratinocyte migration^{38,39}; gain- and loss-of-function experiments have shown that ARHGAP29 positively regulates oral keratinocyte migration in vitro; and a mutant variant of ARHGAP29 identified in orofacial clefting patients is unstable and fails to accelerate cell migration. (Liu et al, in press).”

Liu, H., Busch T., Eliason⁴, S., Anand, D., Bullard, S., Gowans, L.J.J., Nidey, N., Saadi, I., Lachke, S.A., Zhu⁹, Y., Adeyemo, A., Amendt, B., Tony Roscioli, R., **Cornell, R.A.**, Murray, J., Butali, A. (2016) Exome Sequencing Confirms the Role of *ARHGAP29* in Mendelian Orofacial Clefting. **Birth Defects Research Part A: Clinical and Molecular Teratology** (in press)

Reviewers' Comments:

Reviewer #1 (Remarks to the Author)

The present manuscript addresses an important and burning question in complex genetics: how to translate genetic findings into biology. In traits such as orofacial clefting, which occur early in embryonic development, the answer is even more challenging as follow-up studies are hampered by inaccessibility of relevant tissues or human samples. The present study suggests a comprehensive experimental pipeline that dissects functionally relevant risk variants from those that are in high linkage disequilibrium but do not contribute at biological level. This pipeline is likely to be used for the follow-up of different risk loci in clefting and other developmental traits/birth defects. The manuscript lays out an impressive amount of work, and my previous comments on the first version have been successfully addressed. I only have one last minor comment:

Results, line 150f: In the last sentence, the authors state that "we chose to pursue the three whose effect was in the same direction,...". Could the authors please clarify what is meant by "the same direction"? It is unclear to the reader whether it relates to the same genetic effect (i.e., major allele as risk allele), or to the functional effect detected in the assay.

Robert A. Cornell, Professor
Department of Anatomy and Cell Biology
University of Iowa College of Medicine
1-400D Bowen Science Building
Iowa City, Iowa 52242-1109
319-335-7753 Tel
319-335-7198 Fax

Response to Referees Letter

Reviewer #1 (Remarks to the Author):

The present manuscript addresses an important and burning question in complex genetics: how to translate genetic findings into biology. In traits such as orofacial clefting, which occur early in embryonic development, the answer is even more challenging as follow-up studies are hampered by inaccessibility of relevant tissues or human samples. The present study suggests a comprehensive experimental pipeline that dissects functionally relevant risk variants from those that are in high linkage disequilibrium but do not contribute at biological level. This pipeline is likely to be used for the follow-up of different risk loci in clefting and other developmental traits/birth defects. The manuscript lays out an impressive amount of work, and my previous comments on the first version have been successfully addressed. I only have one last minor comment:

Results, line 150f: In the last sentence, the authors state that “we chose to pursue the three whose effect was in the same direction, ...”. Could the authors please clarify what is ment by “ the same direction”? It is unclear to the reader whether it relates to the same genetic effect (i.e., major allele as risk allele), or to the functional effect detected in the assay.

- *We agree with the reviewer’s comment that this saying is ambiguous. “the same direction” in line 150f means “risk allele that decrease enhancer activity in vitro”, and we made changes accordingly in the revised manuscript.*